# ESBL and AmpC β-Lactamases in Clinical Strains of *Escherichia coli* from Serra da Estrela, Portugal

**DOI:** 10.3390/medicina55060272

**Published:** 2019-06-12

**Authors:** Cátia Oliveira, Paula Amador, Cristina Prudêncio, Cândida T Tomaz, Paulo Tavares-Ratado, Rúben Fernandes

**Affiliations:** 1School of Health, Polytechnic of Porto, 4200 Porto, Portugal; catia.oliveira@ulsguarda.min-saude.pt (C.O.); cps@ess.ipp.pt (C.P.); 2Faculty of Biology, University of Vigo, 36310 Vigo, Spain; 3Sousa Martins Hospital, ULS Guarda, 6300 Guarda, Portugal; ptavares@ubi.pt; 4CERNAS—Research Centre for Natural Resources, Environment and Society, College of Agriculture, Polytechnic of Coimbra, 3045 Coimbra, Portugal; paula_amador@esac.pt; 5i3S—Instituto de Inovação e Investigação em Saúde, University of Porto, 4200 Porto, Portugal; 6CICS-UBI—Health Sciences Research Centre, University of Beira Interior, 6201 Covilhã, Portugal; ctomaz@ubi.pt

**Keywords:** β-lactamases, ESBL, AmpC, *Escherichia coli*, resistance mechanisms, Serra da Estrela

## Abstract

*Background and Objectives*: Given the considerable spatial, temporal, and ecological factors, heterogeneity, which affects emergency response, persistence, and dissemination of genetic determinants that confer microorganisms their resistance to antibiotics, several authors claim that antibiotics’ resistance must be perceived as an ecological problem. The aim of this study was to determine the prevalence of broad-spectrum *bla* genes, not only Extended-spectrum β-lactamases (ESBL) but also AmpC-types, in clinical strains of *Escherichia coli* isolated from Portugal (in the highest region of the country, Serra da Estrela) to disclose susceptibility profiles among different genotypes, and to compare the distribution of *bla* genes expressing broad-spectrum enzymes. *Materials and Methods:* Clinical strains of *Escherichia coli* presenting resistance to third generation (3G) cephalosporins and susceptibility to inhibition by clavulanic acid were studied by means of phenotypic and molecular profiling techniques for encoding β-lactamases genes. *Results:* Strains were mainly isolated from hospital populations (97%). Molecular analysis enabled the detection of 49 *bla* genes, in which 55% (27/49) were identified as *bla*_OXA-1-*like*_, 33% (16/49) as *bla*_CTX-M-group-1_, 10% (5/49) as *bla*_TEM_, and 2% (1/49) were identified as genes *bla*_CIT_ (AmpC). Among all *bla*_OXA-1-*like*_ detected, about 59% of strains expressed at least another *bla* gene. Co-production of β-lactamases was observed in 40% of strains, with the co-production of CTX-M group 1 and OXA-1-*like* occurring as the most frequent. *Conclusions:* This is the first study using microorganisms isolated from native people from the highest Portuguese mountain regions, showing an unprecedent high prevalence of genes *bla*_OXA-1-*like*_ in this country.

## 1. Introduction

The excessive use of antibiotics in recent years has become an emerging problem across many sectors—public health, economic, and social [1]. Associated with large-scale population movements and the development of industrialization, the current excessive use of antibiotics is the major cause of strains developing mechanisms, consequently causing increased morbidity, mortality, and therapeutic costs [2,3,4,5].

Extended-spectrum β-lactamases (ESBLs) have been widely described and their epidemiology and geographic prevalence is the subject of continuous study [6]. Known for their resistance to third generation (3G) cephalosporins, inhibition by clavulanic acid, and plasmid encoding, ESBLs represent a set of various types of enzymes that differ in origin, amino acid sequence, or hydrolytic activity on β-lactams [1]. The most widespread ESBL variants, especially in strains of *Escherichia coli*, are the TEM-, SHV-, CTX-M-, and OXA-types [7,8,9].

TEM was first discover in 1963 and named after Greek patient, Temionera. Since then, genetic variations of TEM have been described. The first TEM enzyme was named as TEM-1, and nowadays there are over 230 different-TEMs [10]. These enzymes differ in genetic nucleotide missense polymorphisms, thus altering amino acid sequence (Figure 1) and kinetic properties [11].

During the 1990 decade, the TEM- and SHV-type variants were the most prevalent worldwide, with the exception of South America, where the CTX-M-2 was dominant. In that decade, the spread of the TEM- and SHV-type ESBLs were taking place, mostly by endemic clones, with *Klebsiella pneumoniae* being the main carrier of *bla*_ESBL_ genes [8]. The ESBL-producing strains were essentially isolated in healthcare-associated infections (HAI), with high frequency in Intensive Care Units (ICU) and low incidence in community-acquired infections (CAI) [12].

The first epidemiological change began with the spread of the CTX-M-type ESBL [13]. With an extended hydrolytic action and using *E. coli* as a carrier, this type of ESBL became more frequently noted in several parts of the world and in different settings—both HAI and CAI [14]. However, in the middle of the century there was a drastic change in the epidemiology of ESBL-resistance as a consequence of these variants [2].

Throughout this decade, the CTX-M-type ESBLs were the most prevalent β-lactamases in most European, Asian, and South American countries, and in some cases they are even considered endemic [14,15,16]. Despite this wide distribution, mostly due to the CTX-M-15 variant, other variants are locally amplified, such as CTX-M-9 and CTX-M-14 in Spain, CTX-M-1 in Italy, CTX-M-5 in Belarus and Russia, and CTX-M-3 in Eastern Countries [2,7]

OXA β-lactamases were described for the first time in a strain of *Pseudomonas aeruginosa* and showed a low hydrolytic activity on 3G cephalosporins and weak inhibition by clavulanic acid, thus presenting almost no clinical relevance [17]. However, occasional mutations in *bla*_OXA_ genes have caused variants of these enzymes with ESBL phenotypes, extending their spectrum of activity but maintaining their weak inhibition to clavulanic acid. OXA-1 and OXA-1-*like* (OXA-1, OXA-4, OXA-30) variants are examples of these enzymes [18]. Despite more than 220 OXA β-lactamases having been described [10], not all present the ESBL phenotype.

The emerging spread of ESBL-producing strains is currently associated with the production of AmpC β-lactamases. These enzymes are cephalosporinases with inductive activity belonging to the Ambler class C [19,20]. With hydrolytic activity on penicillin and 3G cephalosporins and susceptibility to fourth generation (4G) cephalosporins (cefepime and cefpirome), AmpCs differ from the ESBLs in their hydrolytic action on cephamycins (cefoxitin and cefotetan) and non-susceptibility to clavulanic acid. In contrast to ESBLs, the AmpCs enzymes are only inhibited by cloxacillin or boronic acid [21,22,23].

Originally described as chromosomal β-lactamases, their plasmid encoding is now recognized [24]. AmpCs that are plasmid-mediated are a derivation from the chromosomal enzymes and can co-exist in chromosomal positive strains, such as *E. coli*, increasing their expression [25]. CMY-1 was discovered in South Korea (1989) and was the first plasmid-mediated AmpC [22]. Since then, many different types of plasmid-mediated AmpCs (CIT, DHA, MOX, EBC, FOX, and ACC) have been described, with the CMY-2 enzyme (CIT-type AmpC) being considered the most widespread variant [23,25,26].

The infections caused by ESBL- and AmpC-producing-strains were initially exclusive to HAI, but now they are also found in CAI [14].

Another reason for the resistance to β-lactams and to the emerging spread lies in the phenomenon of co-production of β-lactamases. The ability to produce different variants or types of β-lactamases gives the bacteria an extended resistance profile. This fact, along with the difficulty in detecting these phenomena by phenotyping techniques usually applied in hospital laboratories, decreases the viability of the commonly used antibiotherapies, and therefore, allows for the spread of these enzymes [27,28,29].

In Portugal, several studies have been conducted in order to describe the epidemiology, prevalence, and emerging spread of variants, in particular the β-lactamase-producing strains with possibility of resistance. Several Portuguese regions, such as the north and central, have been targeted by these studies [7,9,18,30,31,32,33]. However, due to constant epidemiological changes and to the absence of studies in other regions, this purpose has not yet been completely achieved.

This work aims to assess the prevalence of ESBLs and AmpCs in clinical isolates of *E. coli* in the region of Serra da Estrela, Portugal. Additionally, it intends to determine the most prevalent *bla* gene in this region; to provide evidence for the co-production phenomena, as well as its frequency and prevalence, and to compare antibiotic susceptibility profiles among different genotypes and to understand the distribution of *bla* genes in different settings, nosocomial (HAI), and community environments (CAI).

## 2. Materials and Methods

### 2.1. Bacterial Strains

In the present study, strains of *E. coli* (n = 38) with resistance or low susceptibility to 3G cephalosporins detected in the Service of Clinical Pathology, Sousa Martins Hospital, ULS Guarda, Portugal, from July 2010 to May 2011, were analyzed. These strains were isolated from different biological specimens derived from different settings—HAI and CAI.

### 2.2. Phenotype Identification and Antibiotics Susceptibility

Bacterial identification and antibiotic susceptibility profiles were determined by the Vitek^®^2 (bioMérieux®, Marcy-l’Étoile, France) automated system, using the ID GN card (bioMérieux®, Marcy-l’Étoile, France) for identification of Gram-negative susceptibility, and then the AST N151 card (bioMérieux®, Marcy-l’Étoile, France) for Gram-negative susceptibility. The susceptibility to cefoxitin (not evaluated by the AST N151 card) was analyzed using the disk diffusion method (Kirby-Bauer method), according to the standards of the Clinical and Laboratory Standards Institute (CLSI) for the determination of minimum inhibitory concentrations (MIC) [34]. Briefly, the susceptibility to this cephamycin was tested with a cefoxitin disk (30 μg) and confirmed whenever a halo diameter higher than 18 mm was observed. This criterion is mentioned by several authors as a screening method for the detection of AmpC-type β-lactamase producing strains [35].

The production of ESBLs was evaluated by E-test (bioMérieux^®^, France): E-test cefotaxime and cefotaxime/clavulanic acid (CT/CTL) and the E-test ceftazidime and ceftazidime/clavulanic acid (TZ/TZL). The production of ESBLs were determined as positive whenever the MIC ratio was equal to or higher than 8 (CT/CTL or TZ/TZL ≥ 8), or if there was a *phantom zone* or an ellipse deformation, according to the manufacturer’s instructions. Similarly, the screening for AmpC β-lactamases production was determined by E-test (bioMérieux^®^, Marcy-l’Étoile, France): cefotetan and cefotetan/cloxacillin (CN/CNI).

### 2.3. Molecular Characterization of bla Genes

A set of three multiplex Polymerase Chain Reaction (PCR) assays and a simple PCR assay was used for the amplification of the most frequent *bla* genes in *E. coli* strains, as described by Dallenne et al. (2010) and represented in Table 1 [36].

The reaction mixture was prepared with 1μL (2ng) of DNA in a total 25μL of *master mix* with a 1xPCR buffer (10 mM Tris-HCl, pH 8.3/50 mM KCl), 200μM of each dNTP (deoxyribonucleotide triphosphate mix), 1.5 mM MgCl_2_, a variable concentration of specific *primers*, and 0.5U of *Taq polymerase*. The amplification occurred under the following conditions: initial denaturation at 94 °C for 10 min; 30 cycles at 94 °C for 40 s, at 60 °C for 40 s, at 72 °C for 1 min, final elongation at 72 °C for 7 min, and holding temperature at 4 °C.

Amplified products were evaluated by 2% agarose gel containing SYBR green (SYBR® Green I, SigmaAldrich, Spain) electrophoresis, which ran at 80V for 4 h in 1xTAE [4.844 g/L of 2-Amino-2-hydroxymethyl-propane-1,3-diol (TRIS base), 1.21 ml/L of of glacial acetic acid, 0.372 g/L of 2,2’,2’’,2’’’-(Ethane-1,2-diyldinitrilo)tetraacetic acid (EDTA) sodium salt, pH 8.0]. The visualization of the fragments was carried out in the UV transilluminator, where the molecular weight was inferred through a molecular weight marker (100 bp), allowing for the identification of the enzyme.

## 3. Results

Phenotypic susceptibility to antimicrobials, as well as molecular characterization of *E. coli* isolated from the region of Serra da Estrela, Portugal, are shown in Table 1.

### 3.1. Bacterial Strains

In the present study, 38 strains of *E. coli* were harvested, isolated, and studied. About 97% (37/38) of strains were isolated from HAI, supporting the literature that highlights the hospital setting as prevailing when referring to multi-resistant bacteria [14]. Only 3% were isolated from CAI. Strains were isolated mainly from urinary infections with a frequency of 68% (26/38), 18% (7/38) from soft tissue and skin infections, 10% (4/38) from respiratory infections, and 3% (1/38) from the bloodstream.

### 3.2. Phenotype Identification and Antibiotic Susceptibility

The susceptibility to antimicrobial agents was carried out on Vitek^®^2 (bioMérieux^®^) and confirmed an extended profile of resistance to several antibiotics. Besides ampicillin (inclusion criterion), all of the strains presented resistance to other β-lactams, in particular cephalotin and cefuroxime (2^nd^G cephalosporins), cefotaxime, and ceftazidime (3G cephalosporins), and all were susceptible to ertapenem and meropenem, which are carbapenems (Table 2).

Regarding non-β-lactams, antibiotics from all of the strains in the study were susceptible to tigecycline, a tetracycline. Also, 97% (37/38) of the strains evidenced resistance to fluoroquinolones (ciprofloxacin and levofloxacin), 71% (27/38) to aminoglycosides (gentamicin and tobramycin), and only 3% (1/38) to amikacin (data not shown). Of all strains studied, 55% (21/38) were resistant to the combination of trimethoprim/sulfamethoxazole.

Phenotypic production of ESBLs was confirmed by E-test for 97% (37/38) of the strains. The remaining 3% (1/38) correspond to the strain that showed inconclusive results in both E-tests.

Cefoxitin (cephamycin) was used as a screening test for AmpCs producers in combination with E-test. Concerning this antibiotic, 37% of the strains (14/38) were resistant to cefoxitin, while AmpC E-test confirmation gave only one positive result.

### 3.3. Molecular Characterization of bla Genes

The evaluation of *bla* genes was performed by multiplex PCR for TEM, SHV and OXA (TSO), CTX-M, and plasmid AmpC. In the 38 strains, 49 *bla* genes were detected, which indicates the co-production of several *bla* genes within the same strain.

The detection of *bla* genes was confirmed in 87% (33/38) of the strains under study. The remaining 13% correspond to isolates that did not amplify any of the studied *bla* genes (Figure 2).

Concerning the 37 ESBLs producers detected by E-test, 32 were confirmed by PCR, indicating its high sensitivity (97%). Among the 49 *bla* genes detected, 55% (27/49) were *bla*_OXA-1-*like*_, 33% (16/49) were *bla*_CTX-M group 1_, 10% (5/49) were identified as *bla*_TEM_, and only one gene, or 2% (1/49), was identified as *bla*_CIT_.

The production of OXA-1-*like*-type ESBL was found in 71% (27/38) of the isolates, and 44% of the time (12/27), at least another *bla* gene was detected in the same microorganism. Conversely, CTX-M-1 group ESBL was found in 42% (13/38) of the strains, but up to 81% of the time (13/16) in co-production with other types of *bla* genes.

Among the many *bla* gene combinations identified in the present study, CTX-M-1 group and OXA-1-*like*-type ESBLs were predominant, appearing up to 26% of the time (10/38).

The co-production of *bla*_TEM_ and *bla*_CIT_ were detected in the same strain E4 (Figure 3). Although this strain gave inconclusive E-test results for ESBL, the result obtained from E-test for AmpC was positive and confirmed by PCR.

## 4. Discussion

The high prevalence of ESBLs and plasmid-mediated AmpCs strains of *Enterobacteriaceae* is a major concern worldwide [26]. In this study, 38 *E. coli* strains with reduced susceptibility and high resistance to 3G cephalosporins were evaluated.

About 97% (37/38) of strains isolated in the present study came from health care facilities. According to the literature, hospitals are still a setting in which multi-resistant bacteria are predominant [12]. As mentioned, only one strain was found from a community acquired infection (CAI). Nonetheless, it was found that the patient carrying such a strain was hospitalized afterwards, thus the fact that it could be another healthcare acquired infection (HAI) cannot be excluded.

The resistance to different antibiotics classes has been described in ESBL- and AmpC-producing-strains. Genes conferring resistance to quinolones and aminoglycosides have been extensively reported in the same plasmid harboring *bla* genes [18]. These resistance profiles confine the therapy choices and allow the spread of multi-drug resistant plasmids [17]. In this study, high resistance to fluoroquinolones (97%, 37/38), to aminoglycosides (71%, 27/38), and to the trimethoprim/sulfamethoxazole (55%, 21/38) were also found. These results are, in general, highest when described in previous studies [14,19], however, they are corroborated by other Portuguese studies [7,9,18,37], suggesting a local resistance profile that may be due to the spread of these multi-drug resistance plasmids in Portugal. Besides this extended resistance profile, all strains exhibited susceptibility to tigecycline. This result may be due to the fact that genes encoding to tetracyclines resistance are less frequently described in plasmid-borne ESBLs and AmpCs than other classes of antibiotic encoding genes [8,17,38].

Regarding β-lactams antibiotics, all strains were resistant to all β-lactams antibiotics tested, with the exception of carbapenems, which were suggestive a priori of the presence of ESBL or AmpC enzymes in such isolates [34,39]. In this study, ESBL E-tests offered a positive result for 97% (37/38) of the strains examined, pointing out a phenotype of ESBLs. These results were confirmed by molecular methods in 32 of the 37 strains, indicating the high sensitivity (97%) of E-tests. Therefore, E-tests are recommended by Clinical and Laboratory Standards Institute (CLSI) and the European Committee on Antimicrobial Susceptibility Testing (EUCAST) for phenotypic detection of ESBLs, and they are widely used in clinical laboratories and research studies [17,34,39,40,41].

However, an inconclusive result (1/38) was found using ESBL E-test. According to Mohanty et al. (2010), this inconclusive result may be due to the co-production of β-lactamases, since the production of AmpCs may mask the ESBLs production in phenotypic tests [20].

The susceptibility to cefoxitin was applied as a phenotypic screening method for AmpC-type β-lactamase production, as referred to in several studies [21,22,26,42,43]. Resistance to cefoxitin was verified in 37% (14/38) of the strains. Mohanty et al. (2010) and Oteo et al. (2010) reported that ESBL β-lactamases are unable to hydrolyze cephamycins [20,26]. However, Bradford et al. (2001) showed that resistance to cephamycins, similar to cefoxitin, may be found in the presence of ESBL enzymes, but suggests that the loss of an outer membrane porin protein may be the cause [40]. In 2009, Fernandes et al. reported high levels of cefoxitin resistance within strains expressing ESBLs in Portugal [44].

As mentioned, AmpC E-test used for phenotypic confirmation of AmpC-type β-lactamase production revealed one positive result only. This result was obtained in the same strain (E4) that exhibited an inconclusive result on ESBL E-tests, corroborating the hypothesis of co-production of ESBL and AmpC enzymes. This result was confirmed by molecular methodology for the detection of the *bla*_TEM_ by TSO multiplex PCR and the *bla*_CIT_ was by AmpC multiplex PCR (Figure 2). The co-production of these β-lactamases has been reported in sporadic cases, mainly in strains of *E. coli* and *K. pneumoniae* [19]. The TEM β-lactamases were considered the most widespread ESBLs until the first 5 years of the last decade by many researchers [9,33,40,44], being substituted with the growing group of CTX-M and plasmid-mediated AmpC β-lactamases, mainly the CMY-2 variant [14,38].

Other cefoxitin-resistant strains that presented a negative result on E-test AmpC did not amplify any *bla*_AmpC_ genes. Therefore, AmpC E-test exhibited a better performance as a phenotypic detection method for AmpC β-lactamases with 100% sensitivity and 100% specificity compared to the cefoxitin screening method, which yielded results of 100% sensitivity but only 65% specificity. These results are contrary to other studies [35,45]. Polsfuss et al. (2011) reported that although the cefoxitin screening showed a lower specificity (78.7%), it evidenced a higher sensitivity (97.4%) than AmpC E-test (77.4%), proposing a flow chart for AmpC detection with cefoxitin resistance as a screening method and the cefoxitin/cloxacillin combination as a confirming phenotypic assay [35]. The ambiguous results obtained may be due to the lower number of isolates evaluated in this study, requiring further investigation.

Although the phenotypic methods are frequently applied in the workflow of clinical laboratories, the molecular biology methods remain the gold standard for β-lactamase detection and identification, and allow for epidemiological knowledge [44]. The evaluation of *bla* genes was performed by several multiplex PCR recipes, allowing for the characterization of 87% (33/38) of the strains (Figure 2). In the remaining 13% (5/38), it was not possible to amplify any of the analyzed *bla* genes. The lack of amplification may be explained by the presence of a possibly new enzyme, due to the high rate of mutations *bla* genes [36,46].

The molecular methods allowed the identification of a total of 49 *bla* genes. Approximately 55% (27/49) were *bla*_OXA-1-*like*_, 33% (16/49) were *bla*_CTX-M group 1_, and 10% (5/49) were identified as *bla*_TEM_. The present work demonstrated a predominance of *bla*_OXA-1-*like*_ followed by *bla*_CTX-M group 1_, and a decrease of frequency of *bla*_TEM_ genes. These results were in agreement with those previously reported by other Portuguese studies in concern to the increased occurrence of OXA-1-*like* and CTX-M ESBLs and the decrease of frequency of the SHV variants (not found at all in the present study) [9,18,37]. However, one of main observations of this study is the predominance of *bla*_OXA-1-*like*_ genes at 55% (27/49), which has never been described before in the country. This may come from an epidemiological change or from an endemic phenomenon in the Serra da Estrela region and may require further detailed studies.

The only bla_AmpC_ gene detected was an AmpC-CIT. According to literature, CMY-2 (a CIT variant) is the most frequent AmpC from *E. coli* [26]. In countries such as the United States and Canada, the CIT-type AmpC has been widely reported in recent years [14]. Baudry et al. reported the presence of the CMY-2 variant in 96% of the AmpC-type β-lactamase *E. coli* producers [38]. Also, Woodford et al. (2007) described a frequency of 44% of CIT-type AmpC *E. coli* producers in European countries, such as the United Kingdom and Ireland [47]. In Spain, Oteo et al. (2010) reported the emerging spread of this variant when they found that 71% of the β-lactamases were of the CMY-2-type [26]. The high frequency of CIT-type AmpC worldwide and the geographical proximity between the study country with Spain, where the spread of this variant is emerging, seem to corroborate the discovery of the CIT-producing strain in this study.

Almost half (45%) of the 27 *bla*_OXA-1-*like*_ found were detected in co-production with other enzymes. According to the literature, the production of OXA-1-*like*-type β-lactamases is usually detected in cases of co-production of β-lactamases due to their lower hydrolytic activity [15,18].

The most common combination, found in 26% (10/38) of the strains, was the CTX-M-1 group with OXA-1-*like*-type (Figure 2). This outcome may be supported by the fact that CTX-M-15 (belonging to CTX-M group 1) was commonly found in co-production with OXA-30 (OXA-1-*like* variant) [48]. This association has been described by several authors in different countries, such as India, the United Kingdom, Canada, and Portugal [18].

## 5. Conclusions

The distribution of *E. coli* strains harboring different variants of broad-spectrum and plasmid-mediated β-lactamases, such as ESBL and AmpC, is still a concerning issue in the Portuguese territory. Molecular identification of ESBLs and AmpCs by PCR allows for comprehension of the local epidemiology, but these tests are expensive. Specialized personnel and equipment are required and they are usually implemented for reference or for investigative laboratories. The phenotypic systems applied in this study may be considered a good alternative for molecular detection of ESBL and AmpC β-lactamases. One of the major constraints and criticisms of the present work may be the fact that we do not confirm the identity of the genetic sequence of the amplified genes by means of sequencing methods. Nevertheless, from the point of view of our healthcare ecology, this study improves greatly upon the knowledge, since it is the first time that this population is studied at the genotype level, with a substantial difference having already been found—based only on the categorization of the main enzymes present—comparative to the frequency and type of enzymes that occur in the rest of the country.

In the Serra da Estrela region, a predominance of OXA-1-*like* enzymes were detected, which may suggest an epidemiological shift of *bla* genes dissemination, or the existence of an endemic phenomenon in the Serra da Estrela, Portugal.

## Figures and Tables

**Figure 1 medicina-55-00272-f001:**
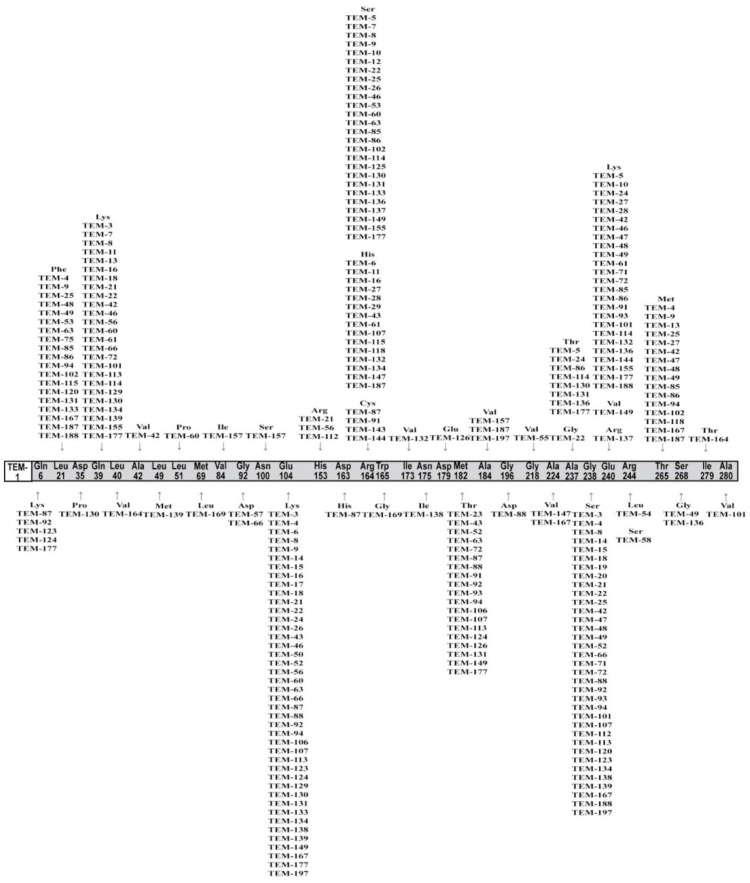
Diagram of the mutations of TEM extended-spectrum β-lactamases, compared to the TEM-1 sequence. Scheme based on available sequences and phenotype in β-Lactamase Data Base (BLDB; permission from authors [11]).

**Figure 2 medicina-55-00272-f002:**
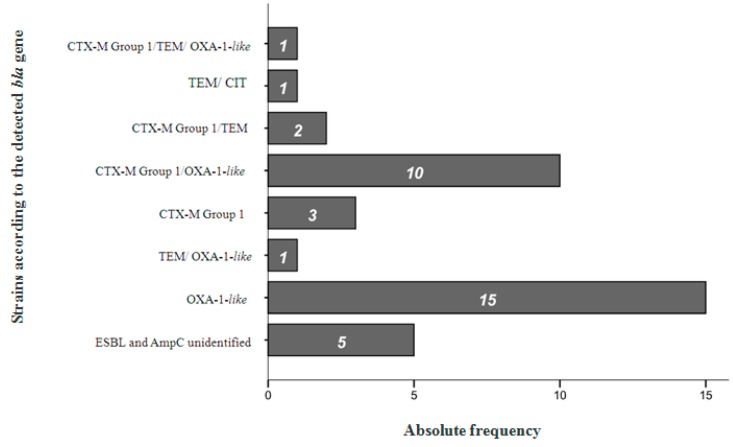
Characterization and frequency of strains sunder study in accordance with the detected *bla* gene.

**Figure 3 medicina-55-00272-f003:**
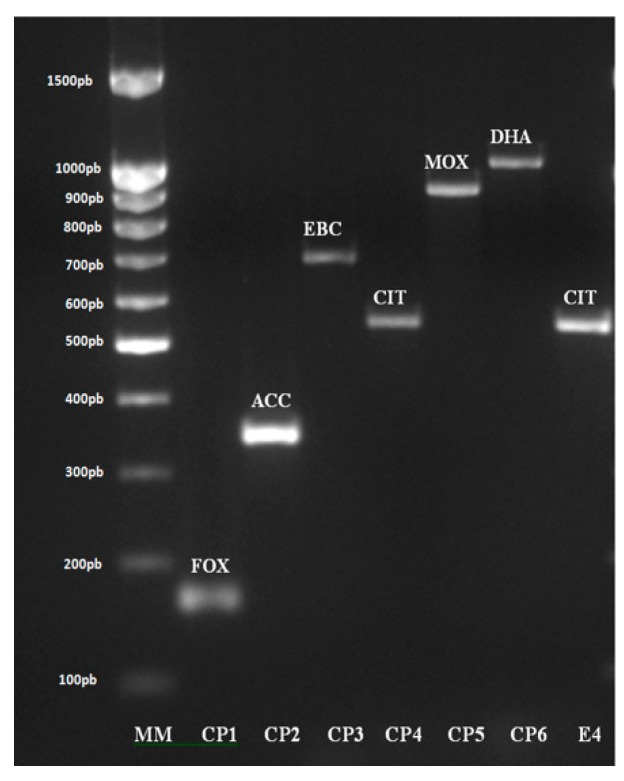
Analysis of the amplified AmpC multiplex PCR products by agarose gel electrophoresis. MM = Molecular weight marker; CP1 = FOX positive control; CP2 = ACC positive control; CP3 = EBC positive control; CP4 = CIT positive control; CP5 = MOX positive control; CP6 = DHA positive control; E4—study strain.

**Table 1 medicina-55-00272-t001:** Sequences of *primers *used for extended-spectrum β-lactamases (ESBL) and AmpC *bla* genes in the present study. Adapted from a previous study [36].

PCR Multiplex	Targeted β-lactamases	Primer’s Name	Sequence (5′-3′)	Product’s Size (bp)
**PCR multiplex TSO** (TEM, SHV, and OXA-1-like)	**TEM variants** (including TEM-1 and TEM-2)	MultiTSO-T_for	CATTTCCGTGTCGCCCTTATTC	800
		MultiTSO-T_rev	CGTTCATCCATAGTTGCCTGAC	
	**SHV variants** (including SHV-1)	MultiTSO-S_for	AGCCGCTTGAGCAAATTAAAC	713
		MultiTSO-S_rev	ATCCCGCAGATAAATCACCAC	
	**OXA-1-like variants** OXA-1, OXA-4 and OXA-30	MultiTSO-O_for	GGCACCAGATTCAACTTTCAAG	564
		MultiTSO-O_rev	GACCCCAAGTTTCCTGTAAGTG	
**PCR multiplex CTX-M** (CTX-M Group 1, CTX-M Group 2, and CTX-M Group 9)	**CTX-M Group 1 variants** (including CTX-M-1, CTX-M-3 and CTX-M-15)	MultiCTX-MGrp1_for	TTAGGAARTGTGCCGCTGYAa	688
		MultiCTX-MGrp1-2_rev	CGATATCGTTGGTGGTRCCATa	
	**CTX-M Group 2 variants**	MultiCTX-MGrp2_for	CGTTAACGGCACGATGAC	404
		MultiCTX-MGrp1-2_rev	CGATATCGTTGGTGGTRCCATa	
	**CTX-M Group 9 variants** (including CTX-M-9 and CTX-M-14)	MultiCTX-MGrp9_for	TCAAGCCTGCCGATCTGGT	561
		MultiCTX-MGrp9_rev	TGATTCTCGCCGCTGAAG	
**PCR CTX-M Group 8/25**	**CTX-M Group 8/25 variants** CTX-M-8, CTX-M-25, CTX-M-26, and CTX-M-39 to CTX-M-41	CTX-MGrp8/25_for	AACRCRCAGACGCTCTACa	326
		CTX-MGrp8/25_rev	TCGAGCCGGAASGTGYATa	
**PCR multiplex AmpC** (ACC, FOX, MOX, DHA, CIT, and EBC)	**ACC variants** ACC-1 and ACC-2	MultiACC_for	CACCTCCAGCGACTTGTTAC	346
		MultiACC_rev	GTTAGCCAGCATCACGATCC	
	**FOX variants** FOX-1 to FOX-5	MultiFOX_for	CTACAGTGCGGGTGGTTT	162
		MultiFOX_rev	CTATTTGCGGCCAGGTGA	
	**MOX variants** MOX-1, MOX-2, CMY-1, CMY-8 to CMY-11, and CMY-19	MultiMOX_for	GCAACAACGACAATCCATCCT	895
		MultiMOX_rev	GGGATAGGCGTAACTCTCCCAA	
	**DHA variants** DHA-1 and DHA-2	MultiDHA_for	TGATGGCACAGCAGGATATTC	997
		MultiDHA_rev	GCTTTGACTCTTTCGGTATTCG	
	**CIT variants** LAT-1 to LAT-3, BIL-1, CMY-2 to CMY-7, CMY-12 to CMY-18, and CMY-21 to CMY-23	MultiCIT_for	CGAAGAGGCAATGACCAGAC	538
		MultiCIT_rev	ACGGACAGGGTTAGGATAGYa	
	**EBC variants** ACT-1 and MIR-1	MultiEBC_for	CGGTAAAGCCGATGTTGCG	683
		MultiEBC_rev	AGCCTAACCCCTGATACA	

^a^ Y = T or C; R = A or G; PCR—Polymerase chain reaction; bp—base pairs.

**Table 2 medicina-55-00272-t002:** Phenotypic and molecular characterization of *Escherichia coli* strains isolated from Serra da Estrela (Portugal) resistant to ampicillin (AMP).

Strain	Source		Antibiogram																										Presumptive Tests								Confirmatory	
																													ESBL				AmpC				Molecular Tests	
																													Etest				FOX		Etest		PCR	
			AMP		AMC		TZP		CXM		CT		TZ		ERT		MEM		GEN		CIP		LVX		TIG		SXT		CT/CTL		TZ/TZL		Diameter		CN/CNI		ESBL	AmpC
			MIC (mg/L)		MIC (mg/L)		MIC (mg/L)		MIC (mg/L)		MIC (mg/L)		MIC (mg/L)		MIC (mg/L)		MIC (mg/L)		MIC (mg/L)		MIC (mg/L)		MIC (mg/L)		MIC (mg/L)		MIC (mg/L)		ratio		ratio		(mm)		ratio			
A1	Urine	HAI	≥32	R	4		≤4		>64	R	>64	R	16	R	≤0,5		≤0,25		≤1		≥4	R	≥8	R	≤0,5		≤20		>8	+	>8	+	27		<8		N.D.	N.D.
A2	Urine	HAI	≥32	R	≥32	R	≥128	R	>64	R	>64	R	≥64	R	≤0,5		≤0,25		≥16	R	≥4	R	≥8	R	≤0,5		40		>8	+	>8	+	23		<8		OXA-1-*like*	N.D.
A3	Respiratory	HAI	≥32	R	16		≤4		>64	R	>64	R	16	R	≤0,5		≤0,25		≤1		≥4	R	≥8	R	1		≤20		>8	+	*P.Z.*	+	24		<8		OXA-1-*like*	N.D.
A4	Urine	HAI	≥32	R	4		≤4		>64	R	>64	R	16	R	≤0,5		≤0,25		≤1		≥4	R	≥8	R	≤0,5		≤20		>8	+	>8	+	25		<8		N.D.	N.D.
A5	Urine	HAI	≥32	R	16		≥128	R	>64	R	>64	R	16	R	≤0,5		≤0,25		≥16	R	≥4	R	≥8	R	≤0,5		≤20		*P.Z.*	+	*P.Z.*	+	24		<8		N.D.	N.D.
A6	Urine	HAI	≥32	R	4		≤4		>64	R	>64	R	16	R	≤0,5		≤0,25		≤1		≥4	R	≥8	R	≤0,5		≤20		*P.Z.*	+	>8	+	24		<8		CTX-M Grp1 + OXA-1-*like*	N.D.
A7	Respiratory	HAI	≥32	R	16		8		>64	R	>64	R	≥64	R	≤0,5		≤0,25		≤1		≥4	R	≥8	R	≤0,5		40		>8	+	>8	+	16	+	<8		CTX-M Grp1	N.D.
A8	Urine	HAI	≥32	R	≥32	R	≥128	R	>64	R	>64	R	≥64	R	≤0,5		≤0,25		≥16	R	≥4	R	≥8	R	≤0,5		40		>8	+	>8	+	24		<8		OXA-1-*like*	N.D.
B1	Urine	HAI	≥32	R	4		≥128		>64	R	>64	R	16	R	≤0,5		≤0,25		≤1		≥4	R	≥8	R	≤0,5		≥320	R	*P.Z.*	+	>8	+	26		<8		OXA-1-*like*	N.D.
B2	Urine	HAI	≥32	R	≥32	R	8		>64	R	>64	R	≥64	R	≤0,5		≤0,25		≤1		≥4	R	≥8	R	≤0,5		≤20		*P.Z.*	+	>8	+	22		<8		N.D.	N.D.
B3	Urine	HAI	≥32	R	≥32	R	≥128	R	>64	R	>64	R	≥64	R	≤0,5		≤0,25		≥16	R	≥4	R	≥8	R	≤0,5		≤20		*P.Z.*	+	*P.Z.*	+	24		<8		CTX-M Grp1 + OXA-1-*like*	N.D.
B4	Respiratory	HAI	≥32	R	16		64		>64	R	>64	R	16	R	≤0,5		≤0,25		≥16	R	≥4	R	≥8	R	≤0,5		≤20		*P.Z.*	+	>8	+	24		<8		OXA-1-*like*	N.D.
B5	Respiratory	HAI	≥32	R	16		64		>64	R	>64	R	16	R	≤0,5		≤0,25		≥16	R	≥4	R	≥8	R	≤0,5		≤20		*P.Z.*	+	*P.Z.*	+	24		<8		N.D.	N.D.
B6	Urine	HAI	≥32	R	16		16		>64	R	>64	R	16	R	≤0,5		≤0,25		≥16	R	≥4	R	≥8	R	≤0,5		≥320	R	*P.Z.*	+	*P.Z.*	+	24		<8		OXA-1-*like*	N.D.
B7	Respiratory	HAI	≥32	R	16		8		>64	R	>64	R	≥64	R	≤0,5		≤0,25		≤1		≥4	R	≥8	R	≤0,5		≤20		*P.Z.*	+	*P.Z.*	+	24		<8		CTX-M Grp1 + TEM	N.D.
B8	Urine	HAI	≥32	R	≥32	R	16	R	>64	R	>64	R	16	R	≤0,5		≤0,25		≥16	R	≥4	R	≥8	R	≤0,5		≥320	R	>8	+	*P.Z.*	+	17	+	<8		OXA-1-*like*	N.D.
B9	Urine	HAI	≥32	R	≥32	R	8		>64	R	>64	R	≥64	R	≤0,5		≤0,25		≥16	R	≥4	R	≥8	R	≤0,5		≥320	R	>8	+	>8	+	26		<8		OXA-1-*like*	N.D.
C1	Urine	HAI	≥32	R	≥32	R	≥128	R	>64	R	>64	R	16	R	≤0,5		≤0,25		≥16	R	≥4	R	≥8	R	≤0,5		≥320	R	>8	+	>8	+	24		<8		CTX-M Grp1 + OXA-1-*like*	N.D.
C2	Urine	HAI	≥32	R	8		≤4		>64	R	>64	R	16	R	≤0,5		≤0,25		≥16	R	≥4	R	≥8	R	≤0,5		≥320	R	>8	+	*P.Z.*	+	24		<8		CTX-M Grp1 + TEM + OXA-1-*like*	N.D.
C3	Urine	HAI	≥32	R	≥32	R	32		>64	R	>64	R	16	R	≤0,5		≤0,25		≥16	R	≥4	R	≥8	R	≤0,5		≥320	R	>8	+	*P.Z.*	+	17	+	<8		OXA-1-*like*	N.D.
C4	Urine	HAI	≥32	R	≥32	R	16		>64	R	>64	R	16	R	≤0,5		≤0,25		≥16	R	≥4	R	≥8	R	≤0,5		≥320	R	>8	+	>8	+	16	+	<8		OXA-1-*like*	N.D.
C5	Urine	HAI	≥32	R	≥32	R	≥128	R	>64	R	>64	R	≥64	R	≤0,5		≤0,25		≥16	R	≥4	R	≥8	R	≤0,5		40		>8	+	*P.Z.*	+	22		<8		OXA-1-*like*	N.D.
C6	Urine	HAI	≥32	R	≥32	R	64		>64	R	>64	R	16	R	≤0,5		≤0,25		≥16	R	≥4	R	≥8	R	≤0,5		≥320	R	*P.Z.*	+	>8	+	17	+	<8		OXA-1-*like*	N.D.
C7	Urine	HAI	≥32	R	16		16		>64	R	>64	R	16	R	≤0,5		≤0,25		≥16	R	≥4	R	≥8	R	≤0,5		≥320	R	*P.Z.*	+	>8	+	24		<8		OXA-1-*like*	N.D.
C8	Skin and soft tissue	HAI	≥32	R	≥32	R	64		>64	R	>64	R	16	R	≤0,5		≤0,25		≥16	R	≥4	R	≥8	R	≤0,5		≥320	R	>8	+	>8	+	15	+	<8		OXA-1-*like*	N.D.
C9	Bloodstream	HAI	≥32	R	16		≤4		>64	R	>64	R	16	R	≤0,5		≤0,25		≤1		≥4	R	≥8	R	≤0,5		≥320	R	*P.Z.*	+	>8	+	24		<8		CTX-M Grp1	N.D.
D1	Urine	HAI	≥32	R	4		≤4		>64	R	>64	R	16	R	≤0,5		≤0,25		≤1		≥4	R	≥8	R	≤0,5		≤20		*P.Z.*	+	>8	+	20		<8		CTX-M Grp1	N.D.
D2	Skin and soft tissue	HAI	≥32	R	16		32		>64	R	>64	R	≥64	R	≤0,5		≤0,25		≥16	R	≥4	R	≥8	R	1		≥320	R	*P.Z.*	+	*P.Z.*	+	15	+	<8		CTX-M Grp1 + OXA-1-*like*	N.D.
D3	Skin and soft tissue	HAI	≥32	R	16		64		>64	R	>64	R	≥64	R	≤0,5		≤0,25		≥16	R	≥4	R	≥8	R	≤0,5		≤20		*P.Z.*	+	*P.Z.*	+	23		<8		TEM + OXA-1-*like*	N.D.
D4	Skin and soft tissue	HAI	≥32	R	≥32	R	16		>64	R	>64	R	16	R	≤0,5		≤0,25		≥16	R	≥4	R	≥8	R	≤0,5		≥320	R	>8	+	*P.Z.*	+	17	+	<8		CTX-M Grp1 + OXA-1-*like*	N.D.
D5	Urine	HAI	≥32	R	≥32	R	≥128	R	>64	R	>64	R	≥64	R	≤0,5		≤0,25		≥16	R	≥4	R	≥8	R	≤0,5		≥320	R	>8	+	>8	+	17	+	<8		CTX-M Grp1 + OXA-1-*like*	N.D.
D6	Urine	HAI	≥32	R	≥32	R	≥128	R	>64	R	>64	R	16	R	≤0,5		≤0,25		≥16	R	≥4	R	≥8	R	≤0,5		≥320	R	*P.Z.*	+	>8	+	24		<8		OXA-1-*like*	N.D.
D7	Skin and soft tissue	HAI	≥32	R	≥32	R	≥128	R	>64	R	>64	R	16	R	≤0,5		≤0,25		≥16	R	≥4	R	≥8	R	≤0,5		≥320	R	*P.Z.*	+	>8	+	17	+	<8		CTX-M Grp1 + OXA-1-*like*	N.D.
D8	Urine	HAI	≥32	R	8		≤4		>64	R	>64	R	16	R	≤0,5		≤0,25		≤1		≥4	R	≥8	R	≤0,5		≥320	R	*P.Z.*	+	*P.Z.*	+	25		<8		CTX-M Grp1 + TEM	N.D.
D9	Urine	CAI	≥32	R	≥32	R	≥128	R	>64	R	>64	R	16	R	≤0,5		≤0,25		≥16	R	≥4	R	≥8	R	≤0,5		≥320	R	>8	+	>8	+	17	+	<8		CTX-M Grp1 + OXA-1-*like*	N.D.
E2	Skin and soft tissue	HAI	≥32	R	≥32	R	64		>64	R	>64	R	≥64	R	≤0,5		≤0,25		≥16	R	≥4	R	≥8	R	≤0,5		≥320	R	>8	+	>8	+	17	+	<8		CTX-M Grp1 + OXA-1-*like*	N.D.
E3	Urine	HAI	≥32	R	16		16		>64	R	>64	R	16	R	≤0,5		≤0,25		≥16	R	≥4	R	≥8	R	≤0,5		≥320	R	*P.Z.*	+	>8	+	15	+	<8		CTX-M Grp1 + OXA-1-*like*	N.D.
E4	Skin and soft tissue	HAI	≥32	R	≥32	R	32		>64	R	8	R	16	R	≤0,5		≤0,25		≥16	R	0,5		1		≤0,5		≤20		Inconclusive		Inconclusive		9	+	>8	+	TEM	CIT

**Legend:** AMC = amoxicillin/clavulanic acid; AMP = ampicillin; CIP = ciprofloxacin; CN-cefotetan; CNI = cefotetan/cloxacillin; CT = cefotaxime; CTL = cefotaxime/clavulanic acid; CXM = cefuroxime; ERT = ertapenem; FOX = cefoxitin; GEN = gentamicin; LVX = levofloxacin; MEM = meropenem; SXT = trimethoprim/sulfamethoxazole; TZ = ceftazidime; TZL = ceftazidime/clavulanic acid; TZP = piperacillin/tazobactam. R or I-resistant or intermediate susceptibility to the antimicrobial agent or combination; N.D. = not determined; + = positive test; HAI = Healthcare-associated infection; CAI = Community-acquired infection.

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
