# Peer review of "ESBL and AmpC β-Lactamases in Clinical Strains of Escherichia coli from Serra da Estrela, Portugal"

_1010-660X, 2019, doi:10.3390/medicina55060272_

Round 1

Reviewer 1 Report

The manuscript by Oliveira et al. investigate the prevalence of the bla genes (ESBL- / AmpC-types) in  clinical strains of E. coli isolated in Serra da Estrela, Portugal; search for co-production mechanisms; compare antimicrobial susceptibility profiles among different genotypes; and compare the distribution of bla genes in different populations (nosocomial and community). However, the manuscript is poorly written and there are so many grammatical mistakes. Proper statistical tools are not used such as the standard deviations/errors  are not  reported. The authors have presented PCR amplification products as evidence for presence of blaTEM and blaCIT , but it would be better if author confirm by sequencing and report variations if present as the sequence can be altered by evolution. The author should also explain about the source of bacteria and present it properly.

Author Response

We like to thank the precious comments. Dissecting reviewer 1 commentaries, we consider 4 main concerns. We will answer point-by-point:

a) "the manuscript is poorly written and there are so many grammatical mistakes.” 

We agree. We have almost re-written the manuscript, which we believe to have improve substantially the quality of the paper.

b) “Proper statistical tools are not used such as the standard deviations/errors are not reported.”

Regarding statistics we are confused about the question. We are not analyzing dispersion measurements to means. Thus, there are no deviations to a central mean to describe, not even errors. We only describe the number of occurrences regarding the number of total events, this is measure is frequency. We prefer to represent it as a relative frequency. We believe to be very explicit, since after every relative frequency (percentage) we mention the number of events by total sample in round brackets.

In that sense were not dealing nor with central limit theorem statistics neither with hypothesis testing. Regarding the last one, our results represent all the events. Thus, we are not comparing samples from a major pool of events and thus, we cannot make hypothesis tests regarding if our sample is representative or not, because we are managing all the events.

c) “The authors have presented PCR amplification products as evidence for presence of blaTEM and blaCIT , but it would be better if author confirm by sequencing and report variations if present as the sequence can be altered by evolution.”

We agree. As we mentioned in the revised conclusions, we know that we be accused of not confirm the identity of the genetic sequence of the amplified genes by means of sequencing methods. Nevertheless, from the point of view of our healthcare ecology, this study brings a great knowledge improvement since it is the first time that this population is studied at the genotype level, having already been found, based on the categorization of the main enzymes present only, a substantial difference, comparatively to the frequency and type of enzymes that occur in the rest of the country. In this sense, we now know that in Serra da Estrela region, there a predominance of OXA-1-like enzymes which may suggest an epidemiological shift of bla genes dissemination or the existence of an endemic phenomenon. We are now motivated to apply for funding to perform this study on a genomic level in this region, which will be a great effort of investment on human resources, time and financial aid.

d) “The author should also explain about the source of bacteria and present it properly.”

We agree. We have introducing a new table for such suggestion.

Reviewer 2 Report

1) Overall, the manuscript needs extensive proofreading with regards to the english language (grammar and missing/misplaced words).

2) Introduction: A visual overview (Figure or Table) with all the different characteristics of the ESBL variants might help the reader to understand this much quicker.

3) The presented data only present select samples. I suggest to show data for all 38 strains in a table(i.e. Diameter or MIC) and put the photographs (e.g. Figure 1) into the supplements.

4) I am wondering whether the authors see differences in the bla gene distribution depending on where the strain is coming from (e.g. urine, wounds, sputum). If so, Figure 2 could be modified accordingly. Which of the strains contained more than 1 bla gene?

5) Figure 3 remains unclear to me. What is the difference between CP1 to CP6? Which strain(s) were tested and what does the reader learn about this experiment. Please explain in more detail.

6) The expression of the different bla genes could be quantified, which might explain differences in the resistance pattern among the 38 strains.

Author Response

We like to thank the precious comments to reviewer 2 commentaries, and we will answer point-by-point all the 6 concerns presented:

1) Overall, the manuscript needs extensive proofreading with regards to the english language (grammar and missing/misplaced words).

We agree. We have almost re-written the manuscript, which we believe to have improve substantially the quality of the paper.

2) Introduction: A visual overview (Figure or Table) with all the different characteristics of the ESBL variants might help the reader to understand this much quicker.

We agree. We have introduced figure 1.

3) The presented data only present select samples. I suggest to show data for all 38 strains in a table(i.e. Diameter or MIC) and put the photographs (e.g. Figure 1) into the supplements.

We agree. We have introduced table 1.

4) I am wondering whether the authors see differences in the bla gene distribution depending on where the strain is coming from (e.g. urine, wounds, sputum). If so, Figure 2 could be modified accordingly. Which of the strains contained more than 1 bla gene?

We have made table 1 in order to answer both questions 3 and 4. We believe that such table has considerably improve reader’s understanding.

5) Figure 3 remains unclear to me. What is the difference between CP1 to CP6? Which strain(s) were tested and what does the reader learn about this experiment. Please explain in more detail.

We agree and have improved legends.

6) The expression of the different bla genes could be quantified, which might explain differences in the resistance pattern among the 38 strains.

That would be must interesting. In fact it would be a major improve to know the expression profile of such enzymes. Also, it has been for long known that also microgenetic environment of the genes conferring resistance to antibiotics have distinct molecular structure, function and behaviours. For instance, some genes were integrated or transposable to regions with different promoter strengths. Also, the same gene could be under the control of different promoters. Moreover, the same gene could be amongst several other genes conferring resistance to other antibiotics, organic solvents and several other biotic and abiotic stressors. Thus, just analyzing gene expression would be somehow reductive. Nevertheless, from the point of view of our healthcare ecology, this study brings a great knowledge improvement since it is the first time that this population is studied at the genotype level, having already been found, based on the categorization of the main enzymes present only, a substantial difference, comparatively to the frequency and type of enzymes that occur in the rest of the country. In this sense, we now know that in Serra da Estrela region, there a predominance of OXA-1-like enzymes which may suggest an epidemiological shift of bla genes dissemination or the existence of an endemic phenomenon. We are now motivated to apply for funding to perform this study on a genomic level in this region, which will be a great effort of investment on human resources, time and financial aid.

Round 2

Reviewer 2 Report

The previous comments were addressed but the ms still could be improved by proof-reading the English.

Medicina EISSN 1010-660X Published by MDPI AG, Basel, Switzerland RSS E-Mail Table of Contents Alert
Back to Top